# Immunomodulation and Biomaterials: Key Players to Repair Volumetric Muscle Loss

**DOI:** 10.3390/cells10082016

**Published:** 2021-08-07

**Authors:** Sonia Kiran, Pankaj Dwivedi, Vijay Kumar, Robert L. Price, Udai P. Singh

**Affiliations:** 1Department of Pharmaceutical Sciences, College of Pharmacy, The University of Tennessee Health Science Center, Memphis, TN 38163, USA; skiran@uthsc.edu (S.K.); vkumar7@uthsc.edu (V.K.); 2Department of Pharmaceutical and Administrative Sciences, University of Health Science and Pharmacy, St. Louis, MO 63110, USA; Pankaj.dwivedi@uhsp.edu; 3Department of Cell and Developmental Biology, University of South Carolina, Columbia, SC 29208, USA; bob.price@uscmed.sc.edu

**Keywords:** VML, immune response, biomaterials, reconstructive therapies

## Abstract

Volumetric muscle loss (VML) is defined as a condition in which a large volume of skeletal muscle is lost due to physical insult. VML often results in a heightened immune response, resulting in significant long-term functional impairment. Estimates indicate that ~250,000 fractures occur in the US alone that involve VML. Currently, there is no active treatment to fully recover or repair muscle loss in VML patients. The health economics burden due to VML is rapidly increasing around the world. Immunologists, developmental biologists, and muscle pathophysiologists are exploring both immune responses and biomaterials to meet this challenging situation. The inflammatory response in muscle injury involves a non-specific inflammatory response at the injured site that is coordination between the immune system, especially macrophages and muscle. The potential role of biomaterials in the regenerative process of skeletal muscle injury is currently an important topic. To this end, cell therapy holds great promise for the regeneration of damaged muscle following VML. However, the delivery of cells into the injured muscle site poses a major challenge as it might cause an adverse immune response or inflammation. To overcome this obstacle, in recent years various biomaterials with diverse physical and chemical nature have been developed and verified for the treatment of various muscle injuries. These biomaterials, with desired tunable physicochemical properties, can be used in combination with stem cells and growth factors to repair VML. In the current review, we focus on how various immune cells, in conjunction with biomaterials, can be used to promote muscle regeneration and, most importantly, suppress VML pathology.

## 1. Introduction

Voluntary muscle or skeletal muscle comprises 40%–50% of the human body, and plays a major role in locomotion, breathing, and posture support [1]. Multiple skeletal muscular diseases adversely affect life and, among these, the most common are muscular dystrophy, [2] cardiomyopathy, [3] myasthenia gravis, [4] poly- and dermatomyositis, and rhabdomyolysis. Interestingly skeletal muscle has an intrinsic ability for regeneration after mild injury. However, volumetric muscle loss [2] that occurs due to surgical or traumatic excision of a portion of skeletal muscle presents an irrevocable condition in humans which leads to sequelae of chronic trauma [5]. VML represents an alarming situation for the U.S. Military Health System, as VML is the main source of disability among service personnel, as well as being important in the civilian population [6,7].

There is a well-known correlation between muscle regeneration and inflammation following acute injury. A mechanistic correlation between muscle inflammation as a result of both innate and adaptive immune dysregulation and regeneration has been recently provided by cellular immunologists, developmental biologists, and muscle pathophysiologists [8]. The in-practice treatment for VML surgery is a follow-up exercise which results in recovery of muscle and its functions. However, the scientific community is struggling with hard to rely on reconstructive therapy for VML. Perhaps most gratifying is that recent approaches have led to new ways to improve inflammatory responses to muscle regeneration both in chronic disease and muscle trauma. Till now, various methodologies, such as activation of immune responses [9], biological scaffolds [10,11], tissue engineering [12], hydrogels [13], and cell transplantation [14], have been utilized to overcome VML. The focus of this review is to highlight the collective role of immune cells, especially macrophages, and various types of biomaterials to overcome the societal burden of treating VML patients and repair of VML.

## 2. Role of Immune Response in VML

Leukocytes are a non-obtrusive element of skeletal muscle that consists of a small portion of intramuscular neutrophils, eosinophils, CD8^+^ cytotoxic T cells, and regulatory T (Treg) cells. It is well known that the larger population of intramuscular leukocytes is composed of monocytes or macrophages [15]. These cells are in the connective tissue sheath that surrounds muscle or blood vessels. The tissue-resident macrophages are also located in a dormant state in healthy muscles. In the case of muscle injury or sudden trauma, the dormant macrophages become rapidly activated and play a role in muscle regeneration or an enhanced wound healing response (Figure 1). As compared to other immune cells, the neutrophils and macrophages are the first line defender in muscle regeneration after any insult or trauma [16]. Following injury, the various immune cells are activated at the spot to remove necrotic cells and to release cytokines [17]. Initially, upon activation, the macrophages secrete tumor necrosis factor α (TNFα), interleukin-1β (IL-1β), and interferon-gamma (IFN-γ), which are also known as pro-inflammatory cytokines, to facilitate cell debris removal. These macrophages further recruit other immune cells at the injured site and they later switch to the anti-inflammatory response that releases anti-inflammatory cytokines to suppress the local inflammatory response and enhance muscle growth [18]. Interestingly, during the phagocytosis of damaged muscle, M1 macrophages reduce TNFα levels, C-C motif chemokine ligand 3 (CCL3), and inducible nitric oxide synthase (iNOS), whereas they enhance expression of CD163, TGFβ1, CD206 by switching to the M2 phenotype. These M2 macrophages are alternatively activated and secrete IL-10 [19,20,21]. The pro-inflammatory macrophages boost the motility of myogenic cells while exerting adverse effects on their differentiation. Furthermore, it has been shown that other major players like hepatocyte growth factor (HGF), insulin-like growth factor-1 (IGF-1), fibroblast growth factor-2 (FGF-2), vascular endothelial growth factor (VEGF), platelet-derived growth factor (PDGF), and Notch signaling in the proliferation of satellite cells plays a significant role in muscle rejuvenation [20,22]. The proliferation in satellite cells results in new dormant satellite cells and myogenic progenitors, which prompt Myf5, MyoD, Mrf4, and myogenin [23,24]. In humans, the affected area contains myogenic progenitors that stimulate macrophages, articulating pro-inflammatory markers like IL-1α, -β, IL-6, TNF-α, and nitric oxide synthase 2 (NOS2) [25,26,27]. However, in vitro analysis reveals that M2 and anti-inflammatory macrophages enhance differentiation of myogenic progenitor [27]. The regenerating muscle contains differentiating myogenic progenitors that are linked with macrophages that protect the anti-inflammatory markers IL4, IL5, IL10, and TGFβ [25,28].

The freshly formed fibers may form an assortment of progressive patterns. The created fibers are forked as newly generated myofibers are fused in an incomplete pattern. Satellite cells also fuse with myofibers under the basal lamina of previously existing fibers [29]. To restore the growth and function of muscles, the mandatory factors are neuro-innervation, myotendinous junctions, and revascularization. If a severe injury occurs, the devascularization causes ischemia, while nerve damage may cause atrophic myofibers. If the injury is left untreated, permanent loss of muscle functions may arise [30,31].

### Inflammation in Immune-Mediated VML

Inflammation is a defensive reaction involving microcirculation and it is initiated after injury or musculoskeletal disease. Understanding the process of skeletal muscle inflammation requires an understanding of key aspects of tissue engineering and regenerative medicine. Inflammatory immune responses have evolved to protect the host from invading pathogens and inflammation, along with the damage or death-associated molecular patterns (DAMPs) generated within the body under diverse inflammatory conditions [32]. Macrophages play a central role in inflammatory responses and during muscle wound healing. Generally, an inflammatory immune response is a highly controlled/regulated process that subsides after clearing the inflammation and the process is called resolution of the inflammation [33]. This resolution process involves various anti-inflammatory mechanisms, including the generation of different anti-inflammatory mediators (cytokines, chemokines, and lipid mediators (resolvins, lipoxins, etc.) and differentiation and recruitment of various immune cells with anti-inflammatory functions mainly through alternatively activated macrophages (AAMs) or macrophage (M2) phenotypes. This is supported by a specialized population of regulatory T cells which also infiltrate the injured muscle and promote the M1 to M2 switch to activate the satellite cells for wound healing [34].

If the inflammatory is not resolved, for example, due to the presence of a highly pathogenic organism that was not cleared, or due to endogenous dysregulation of the immune response during sterile inflammatory conditions, it may be dangerous for the host. Following mild muscle, injury tissues go through a series of processes involving inflammation at the site, repair mechanisms, and remodeling [35]. The inflammatory immune response helps in the necrosis and degradation of skeletal muscle. However, if the loss of skeletal muscle or VML overwhelms the repair mechanism over chronic inflammation at the injured site may occur [36]. Towards this, chronic inflammation enhances the gene expression of many inflammatory modulators [37,38]. Among these, the increase in gene expression of the inflammatory cytokines TGF-β and IL-1β have been reported in VML [38]. To overcome this chronic inflammatory response, physical activity might help delay inflammation and improve the regeneration process [39]. Recently, a group of researchers studying the effect of cyclooxygenase (COX) inhibition on biological scaffold mediated repair of VML, found that COX inhibition blocks macrophage differentiation from the M1 to M2-phenotype [40]. The addition of these non-steroidal anti-inflammatory drugs (NSAIDs) for analgesia may interfere with healing in VML patients [9]. In addition to macrophages, various other immune cells help in muscle regeneration. For example, CD8-T along with macrophages help in the secretion of MCP-1 which ultimately aids in the migration of myeloid-derived suppressor cells Gr1 ^high^ leading to myoblast proliferation and muscle regeneration [41]. Therefore, an immune response that involves macrophages as a major contributor to muscle regeneration and wound healing requires efforts to minimize the chronic immune response in VML.

## 3. Role of Biomaterials in VML

Biomaterial-based therapies for skeletal muscle regeneration in VML use both natural and biosynthetic materials with advanced scaffold assembly techniques to guide cell infiltration and differentiation (Figure 2). Interestingly the latest advances in biomaterials-based therapies paired with a regulated and rigorous evaluation of tissue development have greatly advanced the field of skeletal muscle engineering in the current century [42]. Some of the biomaterials-based therapies for VML are briefly discussed here, and their pros and cons are mentioned in Table 1.

### 3.1. Tissue Engineering in VML

Tissue engineering aims to rejuvenate or repair damaged or completely lost muscle by the combination of cells, growth factors, and biomaterial scaffolds [43]. The development of tissue engineered scaffolds generally aims to provide the mechanical support to the injuries. However, recent advancement has resulted in the tissue engineered scaffolds that are capable to control host cellular functions and functional regeneration. These scaffolds are not only capable of efficiently delivering the encapsulated cells to the site but also, they provide both biophysical and biochemical cues that promote functional skeletal muscle regeneration. Recently, Nutter et al. has synthesized a tissue-engineered skeletal muscle unit and implanted it in a 30% VML tibialis anterior muscle model of rats and observed the generation of muscle fibers at the repair site [44]. Scaffold-free tissue-engineered skeletal muscle units have been designed to analyze the effect of human epidermal growth factor (hEGF) [45]. For example, hEGF serves as the main growth factor for cell proliferation and differentiation. It might aid in the advancement of tissue engineering-based therapies for VML [45]. An experimental model of VML has suggested that an appropriate model should be selected for VML studies considering various factors including volume of muscle removed, the location of the VML injury, and the geometry of the injury [12]. Additionally, different strategies including various biomaterials are under consideration in experimental models designed to enhance the therapeutic outcomes of both cell and growth factor-based approaches in VML, for example, larger animal models may provide a better VML model as compared to small lab animals. In this direction, Rodriguez and his team use the sheep as a craniofacial VML model to highlight the importance of a clinically realistic model [12].

A tissue-engineered scaffold is a template for locally guided and controlled tissue regeneration. Scaffolds may be designed as three-dimensional paradigms consisting of a natural or synthetic substance. These scaffolds provide a synthetic extracellular matrix (ECM) for various cells while imitating the tissue properties of those cells. Various tissue-engineered scaffolds act as an artificial ECM for localized delivery of growth factors and cell populations to the targeted injured skeletal muscle [44,46,47]. These scaffolds are biocompatible and retain the bioactivity of growth factors or proteins. Tissue-engineered scaffolds may be optimized to improve the function of badly damaged skeletal muscles. Biomaterial cargoes have also been developed to integrate synergistic biochemical and biophysical indications that mimic the in vivo satellite cells. These may offer developments in cell reformative potential both in ex vivo development and in vivo rejuvenation [48]. Hence, tissue engineering approaches have significant potential, but challenges such as tissue scaling for size and production speed, etc., need to be addressed before commercial availability of tissue-engineered scaffold for VML become available (Figure 3).

### 3.2. Cell Transplantation in VML

Stem cell therapy has emerged as an encouraging approach for regenerative skeletal muscle repair [49]. The regeneration and homeostasis of skeletal muscle are governed by satellite cells, also termed as muscle stem cells (MuSCs). These cells are present at the boundary of muscle myofibers and are protected by the microenvironment niche in a quiescent state. Following injury, the muscles are activated and differentiated to repair myofibers. The regenerative process comprises numerous cell type interactions, including endothelial cells, immune cells like macrophages, mast cells, smaller populations of T and B cells, and fibro/adipogenic progenitor cells (FAPs) [23,50]. A variety of progenitor and myogenic stem cells are present in regenerating muscles. The MuSCs directly enter the cell cycle process and activate the expression of myogenic cells via myogenic regulatory factor 5 (Myf5) [23] (Figure 4).

Biochemical and physiological phenotypes determine the isolation of MuSCs at different ages and of different embryonic origins. However, it is still a concern that some subpopulations of MuSCs appear at various periods and the relevance of this complexity is not yet known [51,52]. The identification of satellite cells depends upon the expression of the transcription factor paired box 7 (Pax7), as well as anatomical location, but a clear heterogeneity in the population of muscle satellite cells exists. In recent years, various markers, such as CD45, CD11b, Ter119, CD31, and Sca1, have been identified and are used to isolate dormant and activated MuSCs via fluorescence-activated cell sorting (FACS) techniques [53]. Sublaminar satellite cells are sorted due to the transcription factors paired box 3 (Pax3). This factor is only activated in dormant satellite muscle cells, while Myf5 remains inactivated at the injured site. Moreover, all the biomarkers used for the isolation of satellite cells including c-met, CXCR4, and CD34, are familiar to both satellite cells and skeletal muscle tissues [54]. Another important marker which is present in adult satellite cells and during the early development of the muscle laminin receptor is the α7 integrin receptor, which is has a role in the production of ion of the neuromuscular and myotendinous junctions. The ECM and cardiosphere-derived cells (CDCs) are also reported as an ideal candidate for muscle regeneration in VML [55]. Stem cell therapy helps in muscle regeneration, muscular weakness, and muscular dystrophy-associated signs [56,57]. Although extracellular matrix scaffolds are in clinical trials for soft tissue regeneration, cell-based therapy is considered a better approach for VML.

### 3.3. Biological Scaffolds in VML

Biological scaffolds have been successfully designed and applied in clinical applications [58]. It has been found that ideal scaffolds must have a controlled biodegradability for muscle generation, as well as the ability to degrade when skeletal muscle tissues are regenerated. To meet this challenge, chemical crosslinking of natural polymers is employed. These natural polymers have the property to degrade with time when administrated in vivo without any side effects. However, the utilization of numerous biological scaffolds, such as decellularized ECM, is not sufficient for tissue regeneration as they do not have the same function as healthy tissue [59,60,61]. Recently, biomimetic sponges have been reported as more advantageous than decellularized ECM scaffolds [62]. Garg et al. utilize a 20% VML animal model to determine the effect of biomimetic approaches in muscle regeneration. They used enhanced immunomodulatory properties by incorporation of tacrolimus in biomimetic sponges [63]. Tacrolimus is an immunosuppressant that is approved by the FDA [64]. It was observed that biomimetic sponge therapy improved muscle structure and function and reduced inflammation at the injured site in the VML model [64].

To further improve the efficiency of cell transplantation therapy scientists have combined two different approaches, biological scaffolds and cell therapy, to improve muscle regeneration [65]. In addition, the delivery approach and therapeutic cell source are equally important for the success of cell therapy [66]. Biomaterials used in cell transplantation work as artificial niches must mimic the natural environment [67]. This approach is of great clinical interest and improves engraftment and the survival of implanted cells [68].

The innervation of newly formed muscles is often insufficient, but this can be improved. Hence, the outcomes suggest that cell therapy in combination with biological scaffolds and regular exercise is a useful approach in the VML model [69]. Various methods, each with their pros and cons, including both cell-based therapy and biological scaffolds, such as micromolding, microfluidics-based encapsulation, electrospray, and droplet/air are currently in practice. An electrospinning technique can be used for the generation of mature myotubes over microfibers fibrin bundles using tissue engineering. Further stain to the muscles, such as mechanical strain can differentiate myogen to immature myoblasts. Similar to the electrospinning micropatterning can be used to generate skeletal muscle in vitro using micromolded substrates. However, electrospinning and micropatterning generates a 2D model. A 3D scaffold can be generated using 3D bioprinting using bioinks to print bioencapsulated myoblasts. The material which are commonly used to print these 3D scaffolds includes GelMA, alginates, or acrylates. The scaffolds can also be generated using hydrogel micromolding. In which a biocompatible polymer polydimethylsiloxane (PDMS), is commonly used. Cell-based therapy has limitations related to transplantation, harvesting, sorting and expansion, and patient life after transplantation of cells [70]. However, natural materials are superior when compared to synthetic materials due to their biodegradable and biocompatible properties [71]. To sustain muscle regeneration, an ideal biological scaffold should offer prolonged in vivo retention and must be biodegradable after the formation of skeletal muscle tissue. Natural materials that have been used include foam sponges, biofilms, and hydrogels. These materials are cross-linked to improve theirs in situ duration, cell attachment, mechanical properties, and ability to evade the host immune system [72,73].

Muscle regeneration and repair using synthetic scaffolds have also shown promising results. As compared to natural scaffolds, synthetic scaffolds are more advantageous due to preparation reproducibility and flexibility in creating a chemical and physical environment conducive to cell and tissue growth [74]. Various strategies have been used to fabricate scaffolds, such as polymers, including polyethylene glycol (PEG), polylactic acid (PLA), and polydimethylsiloxane (PDMS), coated with adhesion peptides [75,76].

A combination of natural and synthetic scaffolds results in the formation of hybrid scaffolds. These scaffolds act as a synergistic pattern due to their complementary nature. The enhancements in bioactivity due to the presence of natural components stimulate the muscle regeneration process while synthetic components support the physical and mechanical properties. Biological scaffolds in the combination of muscle stimulation boost cell-based therapy in VML, as compared to other approaches employed alone [77,78].

In vivo studies have proven that muscle progenitor cells in combination with ECM scaffolds enhance recovery from VML injury [79]. However, complexity is observed in the case of cellular co-delivery due to the requirement of the patient’s progenitor cells before implantation. To overcome this challenge, minced muscle (MM) autografts that include satellite cells from the stem cell niche may be transplanted [80]. The MM autografts can be collected from healthy muscle tissue and delivered to the injured site. Studies show ~50% improvement in muscle contractile force when MM autografts are transplanted with decellularized muscle scaffolds in the VML model. The MM autografts simultaneously enhance myogenesis and decrease collagen accumulation, which improves the process of regeneration and muscle function after their recovery [75,81]. Therefore, biological scaffolds have been successfully designed and applied in preclinical trials for VML.

### 3.4. Role of Hydrogels in VML Repair

Hydrogels are cross-linked polymeric networks that contain high water content. They can be natural or synthetic in origin and have been used in numerous applications including tissue engineering, immune protection, and drug and cell delivery [82]. The mechanical and structural properties of native tissues can be mimicked by hydrogels [83,84]. Hydrogels that are made from naturally occurring materials can play a vital role in muscle regeneration. They enhance the process of muscle regeneration due to their low inflammatory response and are an important component of ECM [85,86]. The most used natural hydrogels are gelatin, alginate, fibrin, keratin, and collagen. Recently, Pollot and his co-workers studied the mechanical properties of collagen-chitosan fibrin, agarose, collagen, etc., and investigated their effect on skeletal muscle growth [87]. The approximate elastic moduli value of fibrin, collagen, and collagen-chitosan hydrogels is between 2.7 to 3.7 MPa. Agarose hydrogels have an elastic modulus of 87.3 ± 32.6 megapascal pressure units (MPa), hence they are stiffer. The dimension and densities of alginate hydrogel are difficult to predict as alginate hydrogel is malleable [81].

Moreover, it has been hypothesized that fibrin, collagen, and collagen-chitosan hydrogels activate satellite cells if they are implanted at the target site due to their mechanical, biophysical, and biodegradable properties [88]. Semi-synthetic hydrogels, mostly modified by gelatin, are designed by using a naturally occurring polymer [86]. An interesting finding of Kim and his team reported that C2C12 cellular gelatin hydroxyphenylpropionic acid (GHPA) may enhance the proliferation of cells and maturation of myofibers [81]. Another recent finding of hyaluronic acid-based (HyA) based semisynthetic hydrogel for VML reported that applying these hydrogels at the injury site helps in the recovery of injured muscles along with vascularization [89]. Synthetic hydrogels are not considered an ideal approach for skeletal muscle regeneration in VML as compared to natural hydrogels. A hydrogel consisting of one natural and one synthetic polymer is termed a composite hydrogel and is considered the best choice for skeletal muscle regeneration.

One of the most utilized composite hydrogels is PEG. It has been reported that the myogenic and angiogenic activity can be significantly increased using PEG-based hydrogels. PEG-based hydrogels particularly 3D hydrogels are known to rejuvenation of myogenic potential and provides the stiffness. Several attempts have been made to prepare PEG-based hydrogel scaffolds to mimic the natural properties including stiffness and rigidity along with mechanical behavior. Wang et al. prepared the PEG-based hydrogel scaffolds using a dry–wet electrospinning method and incorporated nanofibers yarns in the synthesized PEG-co-poly (glycerol sebacate) composite hydrogel scaffolds to achieve the mechanical properties. In another study it was evident that the PEG incorporation improves the viability of HUVECs cells along with the cellular proliferation and mechanical properties of hydrogels. Hwang et al. applied gelatin PEG-tyramine (GPT) for encapsulation of basic fibroblast growth factor (bFGF) and human adipose-derived stem cells (h-ADSCs) [90]. The bFGF and h-ADSCs incorporated hydrogels were applied to the damaged gastrocnemius muscle. Interestingly, as compared to the experimental group, the twitch muscle contraction was enhanced, and fibrosis reduced in the combined therapy of the bFGF and h-ADSC-hydrogel group. Recently, it has been investigated that injectable self-healing hydrogels are more active, and their biocompatibility and encapsulation efficiency have more patient compliance. Furthermore, self-healing injectable hydrogels are administered as gel formulations [20]. Hence, unnecessary drug load and loss are avoided, which shows promising results in cell therapy. Moreover, if the effective angiogenic factor VEGF is released at the localized pattern via injectable hydrogel, then the function of ischemic muscle tissue is improved [91,92].

Laminin and p38α/β mitogen-activated protein kinase (MAPK) inhibitor-loaded porous hydrogels show positive contributions in the renewal of aged stem cells [93,94]. An interesting cell/hydrogel micromolding methodology was recently applied for designing muscle cells showing 3D structures similar to native tissues [36]. The formulated cell layers were separated from the substrate and could be applied as cell sheets to form multi-layer cell patches or linked to a hydrogel for easy transplantation [95,96]. Hence, hydrogels serve as a suitable candidate for soft tissue regeneration and can help in muscle transplantation in VML.

In addition, the PEG fibrinogen (PF) has attracted many researchers as after the UV exposure it can be transformed into gel. The presence of fibrinogen supports the bioactivity of PF including the cell adhesion on the site of injury in addition to the PEG influencing the material properties. The PF has been clinically approved as a hydrogel which possess both natural and synthetic elements to generate a microenvironment beneficial for proper growth of tissue [97]. PF has been successfully tested in in vitro and in vivo models. The PF encourages the growth of the tissue by providing three-dimensional microenvironment and supports the post-transplant cell survival and myogenic differentiation appropriate for myofiber growth. The gelation properties of PF hydrogel could be modified using the chemical composition. By modifying the chemical properties of PF could support the gelation of PF hydrogel inside a muscle injury [98]. Fuoso et al., revealed that the injectable PF hydrogel significantly enhances the mesoangioblasts differentiation in acute and chronic skeletal-muscle degeneration in addition to the enhances survival of transplanted cells [99].

### 3.5. The Artificial Niche

In muscle regeneration, it is of the utmost importance to understand the mechanism of regeneration in normal tissue. Hence, the main purpose of combining biopolymers, stem cells, and growth factors is to create an artificial niche for muscle regeneration in VML. These synthetic niches follow the natural methods of both differentiation and self-renewal. The local microenvironment of stem cells helps to maintain their identity and regulate their function. The characterization of niches is clear in intestinal crypt stem cells, hematopoietic stem cells, neural stem cells, hair follicle stem cells, and Drosophila germline stem cells [100,101]. The muscle stem cell niche signals are difficult to characterize due to their mechanical, electrical, and chemical properties. The stem cells are located under the basal lamina along with muscle fibers [102,103]. The basal lamina, consisting of collagen, laminin, and proteoglycans, is very important for creating a functional MuSC niche [104]. Moreover, the nourishment of stem cell microvasculature is a vital component of the SC niche and endothelial cells. The finding proves that signals from the host circulation system, muscle fibers, and ECM manage the dormancy, initiation, and proliferation of MuSCs [23,105]. The synthesis of an artificial 3D microenvironment must mimic the natural niche. The microenvironment of MuSCs is polarized in structure and is in the basement membrane and basal lamina. The ideal niche model allows recapitulation of this structure and its amplification in the muscle engineering process to decide the dormant and activation timing of cells [75,106]. In the past, different models have been applied for the in vivo transplantation of artificial niches, such as hydrogels as used to deliver the stem cell to aged, damaged, or diseased tissue sites [67,107]. Moreover, these hydrogel-based biomaterials safely deliver the cells, enhance their viability, and promote the role of endogenous stem cells. These biomaterials also deliver cytokines such as TNF-α, IL-1β, IL-6, IL-8, to enhance the mobilization of endogenous cells, in turn repairing endothelial progenitors and forming blood vessels.

In another model, the biomaterials were applied to boost the function of endogenous stem-cell via local delivery of niche components. These bioactive components can inhibit or stimulate components and promote stem cell growth and their function in niches. It is still a challenging situation to design injectable, multi-component biomaterials which function as a de novo niche when they are administered in vivo [75,105,108]. Together the artificial niches support muscle regeneration and serve as an alternative approach for the repair of VML.

### 3.6. Non-Invasive Therapies for VML

To promote and stimulate muscle regeneration, non-invasive strategies have also been applied. Some of the approaches include light or heat therapy, electrical stimulation, [109], and magnetic stimulation. [110] The resulting stimulation helps in the activation of voluntary muscle following surgery. Moreover, it helps to overcome muscle disuse atrophy when used in combination with nutrients and proteins [111,112,113,114]. A non-invasive strategy known as near-infrared light therapy improves the contractile performance of a damaged muscle [115]. A low level of laser therapy is also effective for the improvement of the volume and morphology of muscles in the gastrocnemius muscle. The combination of laser therapy with plasma having excess platelets enhance muscle regeneration as compared to the use of laser treatment individually [75]. Heat stress in mouse models also enhances muscle regeneration following crush injury. Hence, these non-invasive therapies also play a role in muscle regeneration for VML.

## 4. Conclusions and Future Direction

VML is a worldwide problem that affects the life expectancy of the military, as well as civilian populations. In VML, muscle mass and function are entirely lost, and physiological repair mechanisms of muscles are overwhelmed. Hence, there is the utmost need to design new strategies for skeletal muscle regeneration that take inflammatory, immune response, and biomaterials together in the same direction and act as a two-edged sword for the treatment of VML. The most recent technological advancement for in situ tissue engineering has been accomplished by a combination of various cues incorporated into a single biomaterial. In these settings’ immune cells, especially macrophages, play a very crucial role in tissue regeneration, reduced inflammation, and repair of VML. The use of cell-based therapy or biomimetic scaffolds provides regenerative signals or structural support for the host or transplanted cells through immune modulation. The structure and function of biomaterials with diverse chemical and physical properties have been identified and investigated. In these strategies, biomaterials are either utilized alone or in combination with ex vivo cultured cells or exogenous growth factors. Although the VML challenge is overwhelmed by the utilization of cell-based therapies and biomimetic scaffolds in clinics with positive results, due to the unique immune response of individuals, it creates a hurdle for incorporation of synthetic muscle at the injured site, therefore researchers have a new direction to take the immune response and artificial muscles transplantation in one direction.

## Figures and Tables

**Figure 1 cells-10-02016-f001:**
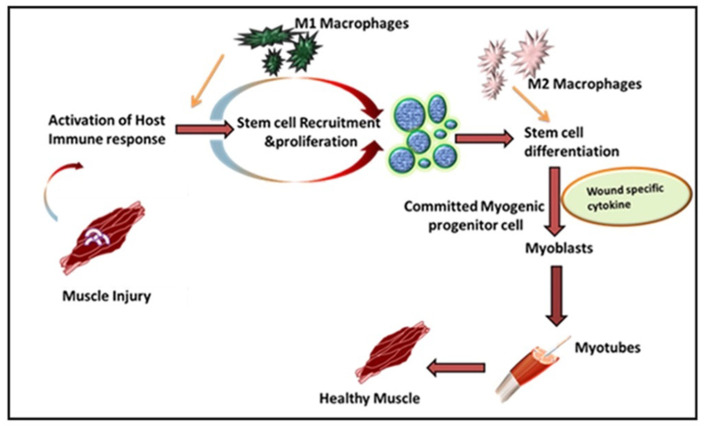
**Immune response in VML**: In mild or severe injury the host immune response is activated. The tissue-resident stem cells are recruited at the site and their proliferation begins. M1 and M2 macrophages are activated which clear the damage and repair the tissue and fibrosis D.

**Figure 2 cells-10-02016-f002:**
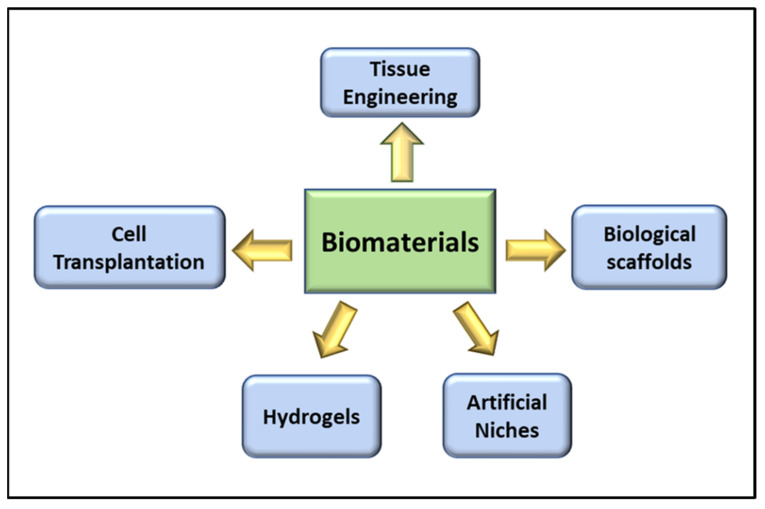
Schematic illustration of current biomaterial approaches for the treatment of VML.

**Figure 3 cells-10-02016-f003:**
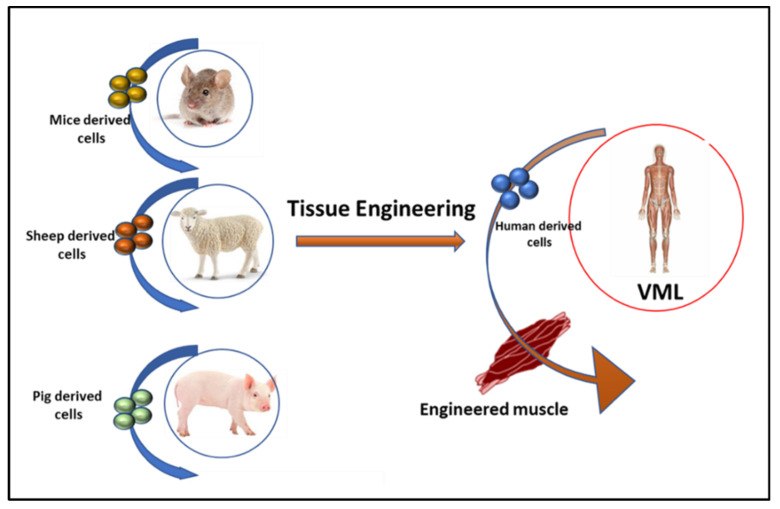
**Tissue engineering for VML**: Tissue engineering techniques help to repair damaged or completely lost muscle in small (lab animals) and large animals including sheep and pigs by the combination of cells, growth factors, and biomaterial scaffolds. The engineered muscle is then transplanted in VML patients.

**Figure 4 cells-10-02016-f004:**
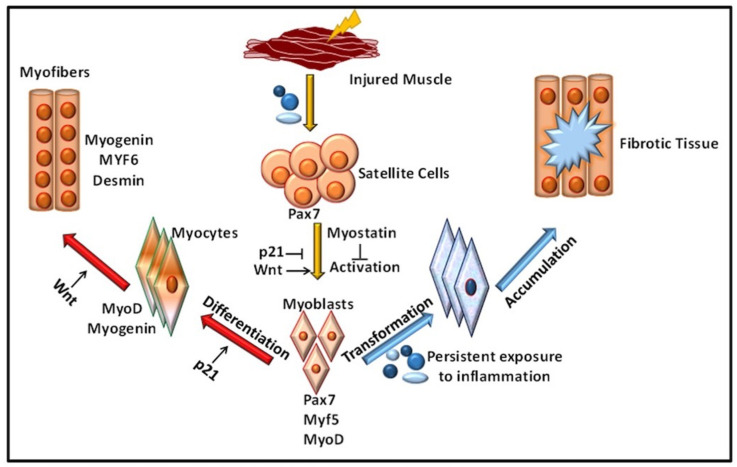
**Cell transplantation in VML**: As a result of the injury satellite cells are differentiated to generate muscle fibers in VML.

**Table 1 cells-10-02016-t001:** Advantages and disadvantages of methodologies used in the therapy of VML.

	Advantages	Disadvantages
Tissue Engineering	Possess biomimetic properties, native-like structural properties and, contain densely packed and uniformly aligned myofibers throughout a relatively large tissue volume. Can be prepared by a simple process to generate micro/nanofibers mimicking the biological environment to promote cellular activities. Material used are biocompatible, biodegradable, viscoelastic and they could be chemically modified Non-toxicity, non-allergenicity, mucoadhesive Range of melting temperature, little inflammatory response Excellent mechanical properties, good biocompatibility Degradation products can be excluded from the body Complex muscle structures in vitro for subsequent implantation, as well as replacement of the missing muscle	Lack of methods to fabricate muscle constructs with native tissue architecture and sufficient force generation capability. Prevent poor survival and integration of tissue constructs into the host muscle. Rapid degradation, water solubility Poor mechanical strength Poor cell attachment Hydrophobicity, slow degradation, Poor mechanical properties, hydrophobicity
Cell Therapy	Cell therapy has been, by far, the most promising approach to treat skeletal muscle injuries in pre-clinical settings Easily isolated from bone marrow High chondrogenic potential Broadly characterized and investigated Homogeneous population Abundance of tissue High yield Low donor tissue morbidity High proliferative rate High chondrogenic potential Low donor site morbidity	Limitations of cell therapy approaches include issues related to autologous harvesting, expansion and sorting protocols, optimal dosage, and viability after transplantation. Demonstrated success in the handful of clinical trials performed so far. Extracting cells from bone marrow is a very painful and invasive procedure Low yield (approximate 1 in 1 × 105 cells in the marrow) Decline in proliferative and differentiation capacity with age Non-homogeneous cell population Limited source of tissue
Biological Scaffolds	Biological scaffolds are used in the clinic on human patients (FDA approved). Promote the regeneration and repair in VML, in part by acting as regenerative templates and modulating healing processes The materials used could be natural, synthetic or hybrid Reinforcement of soft tissues repaired during tendon repair surgery Could be combined with cell-based therapy for better outcome	Should be chemically modified for long in vivo life to sustain muscle regeneration Incomplete alignment of the regenerating tissue with the healthy one. Biomaterials are sometimes rejected by the receiver’s immune system.
Hydrogels	Degrade in the body, leaving behind their biological payload in a process that can enhance the therapeutic process. They provide mild processing conditions and have similarities to natural tissue extracellular matrix. Could be delivered with less invasive approaches Synthetic hydrogels tend to have a relatively low risk of pathogen transfer	Reduced mechanical properties, uncontrollable degradation, and lack of cellular adhesion receptors.

## Data Availability

Not applicable.

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
