# Peer review of "Immunomodulation and Biomaterials: Key Players to Repair Volumetric Muscle Loss"

_cells, 2021, doi:10.3390/cells10082016_

Round 1
Reviewer 1 Report
The paper by Kiran et al reviews means and materials to repair muscle loss due to physical insult. This is in contrast to the muscle loss due to typically inherited muscle disorders. As tremendous research efforts has been put into treating muscle disorders, and less effort has been put into reversing volumetric muscle loss (VML).
The review is well written and well organized and very interesting. I do however have a few comments that needs to be addressed to make some of the points better understandable. The review also needs to make clear the difference in immune response for inherited muscle disorders, in which there is an ongoing degeneration/regeneration cycle, the immune response activating muscle repair in otherwise uncompromized muscle (apart from the VML) and finally which may be most important the immune response that may be modulated pharmacologically.
In line 53-54 various methodologies for overcoming VML are listed, but at this point several of these means little to the reader. some are later excellently explained while others are explained in much less detail. Specific comments to these follows below. An explanatory table listing the types of methodologies may be helpful, explain what biological scaffolding is, especially if there are different kinds, tissue engineering etc. Their use are reviewed in separate sections, but little about what they consist of, materials, structure, mix of synthetics and biological material. This could contain pro's and con's as mentioned below.
In line 138, COX inhibitors are used to block macrophage differentiation from M1 to M2-phenotype, yet it is M2 phenotype that promotes activation of satellite cells for muscle repair/regeneration/wound healing (line 123), while line 79-80 mentions that the M1 phenotype activates muscle progenitor cells in vitro. The authors should explain the apparent inconsistency in this.
It may also be helpful to explain when satellite cells which forms the majority of muscle repair and sidepopulation are activated.
In line 160 a ”tissue-engineered skeletal muscle unit ” is mentioned. How is this defined, what does it consistent of in terms of size/function?
In line 165 it is stated that a 30% VML model in rat is not an ideal model for VML studies. Why not? Is the percentage based on loss of limb or fraction of any given muscle. In the real world there is no such thing as ideal, surgical removal or damage from an explosive device may exceed 30% VML.
In line 173 scaffolds are discussed, but what materials are they made of? Are they meant to eventually be absorbed by the muscle or remain as a non-degradable but biocompatible material?
In line 174, please define a ”three-dimensional paradigm”.
In line 192 and forward, cell transplantation is mentioned. This has been tried both with myoblasts and stem cells for more than 25 years to treat Duchenne muscular dystrophy, yet it has not worked. In Duchenne the treatment at least had the theoretical advantage of an existing structure resembling muscle, while this is not the case with VML. The practical intervention using approach needs to be explained better.
In line 240-41 the beneficial effect of biomimetic sponges are mentioned including that it reduced inflammation at the injured site. Yet, inflammation is part of the process that heals the muscle. How is this explained?
In line 252-55 various methods are mentioned, but without further explanation they really do not mean much to the reader. Could the concept of each be explained briefly.
In line 274 ”Biological scaffolds in the combination of muscle stimulation boost cell-based therapy in VML”. What constitutes muscle stimulation?
How does the sentence in lines 284-85 relate to the preceding and following sentence? When during VML is excess collagen deposited and how can biological scaffolds prevent this?
It would be informative given the authors expansive knowledge about this field, which I think ougth to be promoted, to have a table that lists the pro’s and con’s of each of these methodologies and what combination therapies may have succes (in the authors opinion) and what the authors believe may be succesful approaches in geneal to deal with VML (again, in the authors opinion). Please include the NCT identifier for any methods in current, past or soon-to-be launched clinical trial in any table.
Author Response
Reply to Reviewer Response 1:
The paper by Kiran et al reviews means and materials to repair muscle loss due to physical insult. This contrasts with muscle loss due to typically inherited muscle disorders. As tremendous research efforts have been put into treating muscle disorders, and less effort has been put into reversing volumetric muscle loss (VML).
The review is well written and well organized and very interesting. I do however have a few comments that need to be addressed to make some of the points better understandable. The review also needs to make clear the difference in immune response for inherited muscle disorders, in which there is an ongoing degeneration/regeneration cycle, the immune response activating muscle repair in otherwise uncompromised muscle (apart from the VML), and finally which may be most important the immune response that may be modulated pharmacologically.
Thank you very much for allowing us to revise our manuscript. We were very grateful for the comments and suggestions of the reviewers. During this period, we have carefully made corrections and modifications to improve the quality of the manuscript to meet the journal criteria of impact, innovation, and interest. The importance of the work has been highlighted by improving the paragraphs of the original manuscript. The changes to the original manuscript have been tracked by “yellow” in the revised manuscript. Below please find the point-by-point replies to the reviewers’ comments.
Comment 1: In lines, 53-54 various methodologies for overcoming VML are listed, but at this point, several of these means little to the reader. some are later excellently explained while others are explained in much less detail. Specific comments to these follow below. An explanatory table listing the types of methodologies may be helpful, explain what biological scaffolding is, especially if there are different kinds, tissue engineering, etc. Their use is reviewed in separate sections, but little about what they consist of, materials, structure, mix of synthetics and biological material. This could contain pro's and con's as mentioned below.
Response: As suggested by the reviewer, we have illustrated the methodologies used for VML in the revised manuscript.
Comment 2: In line 138, COX inhibitors are used to block macrophage differentiation from M1 to M2-phenotype, yet it is the M2 phenotype that promotes activation of satellite cells for muscle repair/regeneration/wound healing (line 123), while lines 79-80 mentions that the M1 phenotype activates muscle progenitor cells in vitro. The authors should explain the apparent inconsistency in this.
Response: The sentence has been rewritten in the revised manuscript to overcome any confusion. The M1 macrophages are pro-inflammatory, whereas the M2 macrophages are anti-inflammatory. At the early stage of muscle regeneration, M1 macrophages are the most dominant macrophage type, while, M2 macrophages have been indicated to promote muscle stem cells to differentiate to myotubes, thus promoting the late stage of myogenesis and regeneration.
Comment 3: It may also be helpful to explain when satellite cells that form the majority of muscle repair and side population are activated.
Response: As suggested by the reviewer, we have added the description of satellite cells in muscle repair and activation. In intact muscle, satellite cells are subliminal and mitotically quiescent (G0 phase). Upon exposure to signals from a damaged environment, satellite cells exit their quiescent state and start to proliferate (satellite cell activation).
Comment 4: In line 160 a “tissue-engineered skeletal muscle unit” is mentioned. How is this defined, what does it consist of in terms of size/function?
Response: The cells from the muscle isolation and the 5-mm tissue-engineered bone anchors were used to develop a rolled cylindrical muscle constructs held at length by the bone anchors were referred to as tissue-engineered skeletal muscle units.
Comment 5: In line 165 it is stated that a 30% VML model in the rat is not an ideal model for VML studies. Why not? Is the percentage based on loss of limb or fraction of any given muscle? In the real world, there is no such thing as ideal, surgical removal or damage from an explosive device may exceed 30% VML.
Response: We agree with the reviewer that the 30% VML model could be appropriate for the VML study. In the original manuscript, we have mentioned the summary by the research group, who mentioned the importance of balancing the use of a clinically realistic model. Various variables include the volume of muscle removed, the location of the VML injury, and the geometry of the injury, as these affect both the muscle’s ability to self-regenerate as well as the probability of success of the treatment.
Comment 6: In line 173 scaffolds are discussed, but what materials are they made of? Are they meant to eventually be absorbed by the muscle or remain as a non-degradable but biocompatible material?
Response: The objective of tissue engineering is to allow the body's cells, over time, to eventually replace the implanted scaffold or tissue-engineered construct. Scaffolds are not intended as permanent implants and must therefore be biodegradable to allow cells to produce their extracellular matrix. Typically, three individual groups of biomaterials, ceramics, synthetic polymers, and natural polymers, are used in the fabrication of scaffolds for tissue engineering.
Comment 7: In line 174, please define a “three-dimensional paradigm”.
Response: Here for the three-dimensional paradigm, we refer to the shape designed as a 3D scaffold particularly designed by 3D bioprinting. 3D scaffolds can be used as tissue models replicating the structural complexity of the living tissues. Using patient data, the design of the scaffold could be individualized by preparing a special 3D model with certain porosity or structures for vasculature that is compatible with multiple biomaterials and cells.
Comment 8: In lines 192 and forward, cell transplantation is mentioned. This has been tried both with myoblasts and stem cells for more than 25 years to treat Duchenne muscular dystrophy, yet it has not worked. In Duchenne, the treatment at least had the theoretical advantage of an existing structure resembling muscle, while this is not the case with VML. The practical intervention using approach needs to be explained better.
Response: Extensive damage to skeletal muscle tissue due to VML is beyond the inherent regenerative capacity of the body, and results in permanent functional debilitation. MuSCs and MSCs are the two main cell sources that have been evaluated in preclinical cell therapies in VML rodent models and have demonstrated the capacity for muscle regeneration. MuSCs lose their stemness during in vitro expansion, so the development of a robust in vitro expansion method for MuSCs without losing their potency is necessary. Additionally, more studies are needed to understand the molecular mechanisms behind the differentiation and self-renewal behavior of MuSCs. For the translation of cell-based therapies into the clinic, developing a delivery vehicle to promote cell survival in vivo implantation, and obtaining appropriate protocols for cell expansion and storing in vitro, is required.
Comment 9: In lines 240-41 the beneficial effects of biomimetic sponges are mentioned including that it reduced inflammation at the injured site. Yet, inflammation is part of the process that heals the muscle. How is this explained?
Response: We agree with the reviewer, that inflammation is the first stage in the wound-healing process. The biomimetic sponge treatment improved muscle structure and function while modulating inflammation and limiting the extent of fibrotic tissue deposition
Comment 10: In lines, 252-55 various methods are mentioned, but without further explanation, they do not mean much to the reader. Could the concept of each be explained briefly?
Response: As suggested by the reviewer, we have added the explanation in the revised manuscript.
Comment 11: In line 274 “Biological scaffolds in the combination of muscle stimulation boost cell-based therapy in VML”. What constitutes muscle stimulation?
Response: The stimulus can be from various sources such as exercise, a mechanical or electrical stimulus that can enhance stem cell-based therapy in VML.
Comment 12: How does the sentence in lines 284-85 relate to the preceding and the following sentence? When during VML is excess collagen deposited and how can biological scaffolds prevent this?
Response: The sentence has been corrected for better understanding.
Comment 13: It would be informative given the author's expansive knowledge about this field, which I thought to be promoted, to have a table that lists the pro’s and con’s of each of these methodologies and what combination therapies may have success (in the author's opinion) and what the authors believe may be successful approaches, in general, to deal with VML (again, in the author's opinion). Please include the NCT identifier for any methods in current, past, or soon-to-be-launched clinical trials in any table.
Response: As suggested by the reviewer, we have provided the table for different methodologies used in VML therapy.

Reviewer 2 Report
The review entitled “Immunomodulation and biomaterials: key players to repair volumetric muscle loss”, aimed to describe how the different immune cells combined with various biomaterials, can be employed to recover volumetric muscle loss (VML). The authors conceptualized with a general description of the different arguments, but in my opinion the correlation between immune system and biomaterial is not so evident and sometimes seems that the authors missed the point.
Although the review is clear and well-written, minor punctuations errors should be checked (for example: line 150. In the text seems that there are spacing errors…maybe could be a layout problem, please check (for example line 40, line 111, line 117…..)
However, although the review is attractive and mature, I consider certain things to remain unclear or should be clarified, aiming to increase the review readability and impact
Please, find below a summary of my major and minor points of concern regarding the manuscript:
- This is just a suggestion for the Figures: I don’t know if it is a problem produced after pdf generation and downloading but, the quality of the figures seems not to be optimal. In fact, in the figure1, figure3, figure4 some boxes appear not properly inserted.
- In all the text I suggest to the authors to write the entire name when they refer to the different molecules. Sometimes the full name of the proteins is indicate, sometimes not and it is indicated only the acronymous of the protein.
- Line 70-71: when the authors refer to cytokines in my opinion is not so correct write “cytokines activate and differentiate” because they do not act directly on the satellite cells but they have an indirect effect on the cells.
- Line 87: the sentence in which the authors refer to the activity of the satellite cells should be described much better “which promt…..”
- Line 89-91 and also 93-95: what does it mean when the authors write “ are directly associated” or “that are linked” ? I suggest to reformulate the sentence in order to better explain the influence that macrophages have on myogenic progenitors, to which the authors refer.
- Line 139: NSAIDS I suggest to write “Nonsteroidal anti-inflammatory drugs(NSAIDs)”
- The paragraph “Tissue engineering in VML” in my opinion should be improved considering also the necessity to overcome the muscle fiber organization. There are a lot of groups that have introduced the 3D bioprinting to optimize the artificial muscle production. These are well-documented and mature studies published that treat this argument.
- Line 207-215: When the authors indicate the various antigens used to isolated satellite cells, it should be better if they could also introduce the positivity or negativity for the expression of those markers. Moreover, a7integrin is another important satellite cell marker and it should be introduced.
- Line 216-222: this part of the paragraph in my opinion results few related to the previous part. Moreover, in this paragraph the role of the different muscle cell populations treated by the authors, results not sufficiently contextualize in the VML.
- Line 247: Since muscle regeneration is orchestrated by several muscle resident cell, with or without myogenic properties, I suggest to the authors to reformulate the sentence avoiding to refer only to the satellite cells.
- Line 316: Referring to PEG hydrogels authors could improve the manuscript describing also the application of PEG-Fibrinogen in skeletal muscle tissue engineering. Well-documented studies with this hydrogels have been done by several researchers.
- Line 361: IL1-b ” b” is written in bold.
Author Response
Reply to Reviewer Response 2:
The review entitled “Immunomodulation and biomaterials: key players to repair volumetric muscle loss”, aimed to describe how the different immune cells combined with various biomaterials, can be employed to recover volumetric muscle loss (VML). The authors conceptualized with a general description of the different arguments, but in my opinion, the correlation between the immune system and biomaterial is not so evident, and sometimes seems that the authors missed the point.
Although the review is clear and well-written, minor punctuations errors should be checked (for example line 150. In the text seems that there are spacing errors…maybe could be a layout problem, please check (for example line 40, line 111, line 117…..)
However, although the review is attractive and mature, I consider certain things to remain unclear or should be clarified, aiming to increase the review readability and impact
Please, find below a summary of my major and minor points of concern regarding the manuscript:
We were very grateful for the comments and suggestions of the reviewers. We have carefully made corrections and modifications to improve the quality of the manuscript to meet the journal criteria of impact, innovation, and interest. The importance of the work has been highlighted by improving the paragraphs of the original manuscript. The changes to the original manuscript have been tracked by “yellow” in the revised manuscript. Below please find the point-by-point replies to the reviewers’ comments.
Comment 1: This is just a suggestion for the Figures: I don’t know if it is a problem produced after pdf generation and downloading but, the quality of the figures seems not to be optimal. In fact, in figure1, figure3, figure4 some boxes appear not properly inserted.
Response: As suggested by the reviewer, we have provided the figures with better resolution.
Comment 2: In all the text I suggest to the authors to write the entire name when they refer to the different molecules. Sometimes the full name of the proteins is indicating, sometimes not and it is indicated only the acronymous of the protein.
Response: As suggested we have made the names consistent throughout the entire revised manuscript.
Comment 3: Line 70-71: when the authors refer to cytokines in my opinion is not so correct to write “cytokines activate and differentiate” because they do not act directly on the satellite cells but have an indirect effect on the cells.
Response: As suggested by the reviewer, we have modified the sentence in our revised manuscript.
Comment 4: Line 87: the sentence in which the authors refer to the activity of the satellite cells should be described much better “which prompt….”
Response: As suggested by the reviewer, we have added more descriptions of the activity of the satellite cells in our revised manuscript.
Comment 5: Line 89-91 and also 93-95: what does it mean when the authors write “are directly associated” or “that is linked”? I suggest reformulating the sentence to better explain the influence that macrophages have on myogenic progenitors, to which the authors refer.
Response: The sentence has been corrected for better understanding in the revised manuscript.
Comment 6: Line 139: NSAIDS I suggest writing “Nonsteroidal anti-inflammatory drugs (NSAIDs)”
Response: The illustration of the acronym NSAIDs has been added in the revised manuscript.
Comment 7: The paragraph “Tissue engineering in VML” in my opinion should be improved considering also the necessity to overcome the muscle fiber organization. There are a lot of groups that have introduced 3D bioprinting to optimize artificial muscle production. These are well-documented and mature studies published that treat this argument.
Response: As suggested by the reviewer, we have improved the paragraph on Tissue engineering in VML with recent advancements in 3D bioprinting for artificial muscle production.
Comment 8: Line 207-215: When the authors indicate the various antigens used to isolated satellite cells, it should be better if they could also introduce the positivity or negativity for the expression of those markers. Moreover, a7integrin is another important satellite cell marker and it should be introduced.
Response: In our revised manuscript, we have added the description of the markers and added the detail about a7integrin.
Comment 9: Line 216-222: this part of the paragraph in my opinion results from a few related to the previous part. Moreover, in this paragraph, the role of the different muscle cell populations treated by the authors' results not sufficiently contextualize in the VML.
Response: We have modified the paragraph to make it more connected to the previous part in our revised manuscript.
Comment 10: Line 247: Since muscle regeneration is orchestrated by several muscle resident cells, with or without myogenic properties, I suggest to the authors to reformulate the sentence avoiding referring only to the satellite cells.
Response: As suggested by the reviewer, the sentence has been modified.
Comment 11: Line 316: Referring to PEG hydrogels authors could improve the manuscript also describing the application of PEG-Fibrinogen in skeletal muscle tissue engineering. Well-documented studies with these hydrogels have been done by several researchers.
Response: As suggested by the reviewer, we have described the application of PEG-Fibrinogen in skeletal muscle tissue engineering in our revised manuscript.
Comment 12: Line 361: IL1-b ” b” is written in bold.
Response: The font of IL1-b has been made normal from bold.

Round 2
Reviewer 2 Report
I appreciated the good work that the authors have done to improve the manuscript.
I have only few suggestions:
- check the figures organization again, because there are in my opinion some thinks that should be revised. For example: Figure 2: "Biological scaffolds" box is overlaying with the black box of the figure; In Figure 3: also in this figure check some overlaying
- Line387: Thanks to the authors for the better explanation of the part concerning the PEG hydrogel. Nevertheless, the authors t described the results obtained with other PEG-based Hydrogels and not those with PEG-Fibrinogen. (I appreciate this part and my punctualization is just to refer my previous request)
Author Response
Reply to Reviewer 2 Response:
Comment 1: Check the figures organization again, because there are in my opinion some thinks that should be revised. For example: Figure 2: "Biological scaffolds" box is overlaying with the black box of the figure; In Figure 3: also, in this figure check some overlaying.
Response: As suggested by the reviewer, we have provided the revised figure with no overlaying of the borders.
Comment 2: Line387: Thanks to the authors for the better explanation of the part concerning the PEG hydrogel. Nevertheless, the authors described the results obtained with other PEG-based Hydrogels and not those with PEG-Fibrinogen. (I appreciate this part and my punctualization is just to refer my previous request).
Response: As suggested by the reviewer, we have added the role of PEG-Fibrinogen in VML in our current revised manuscript.